# Natural Rubber Composites Using Hydrothermally Carbonized Hardwood Waste Biomass as a Partial Reinforcing Filler- Part I: Structure, Morphology, and Rheological Effects during Vulcanization

**DOI:** 10.3390/polym15051176

**Published:** 2023-02-26

**Authors:** Jelena Lubura, Libor Kobera, Sabina Abbrent, Ewa Pavlova, Beata Strachota, Oskar Bera, Jelena Pavličević, Bojana Ikonić, Predrag Kojić, Adam Strachota

**Affiliations:** 1Faculty of Technology Novi Sad, University of Novi Sad, Bulevar cara Lazara 1, 21000 Novi Sad, Serbia; 2Institute of Macromolecular Chemistry, Czech Academy of Sciences, Heyrovskeho nam. 2, CZ-162 00 Praha, Czech Republic

**Keywords:** natural rubber composites, bio-sourced raw materials, hydrochar, hydrothermal carbonization, carbon black, vulcanization chemistry

## Abstract

A new generation biomass-based filler for natural rubber, ‘hydrochar’ (HC), was obtained by hydrothermal carbonization of hardwood waste (sawdust). It was intended as a potential partial replacement for the traditional carbon black (CB) filler. The HC particles were found (TEM) to be much larger (and less regular) than CB: 0.5–3 µm vs. 30–60 nm, but the specific surface areas were relatively close to each other (HC: 21.4 m^2^/g vs. CB: 77.8 m^2^/g), indicating a considerable porosity of HC. The carbon content of HC was 71%, up from 46% in sawdust feed. FTIR and ^13^C-NMR analyses indicated that HC preserved its organic character, but it strongly differs from both lignin and cellulose. Experimental rubber nanocomposites were prepared, in which the content of the combined fillers was set at 50 phr (31 wt.%), while the HC/CB ratios were varied between 40/10 and 0/50. Morphology investigations proved a fairly even distribution of HC and CB, as well as the disappearance of bubbles after vulcanization. Vulcanization rheology tests demonstrated that the HC filler does not hinder the process, but it significantly influences vulcanization chemistry, canceling scorch time on one hand and slowing down the reaction on the other. Generally, the results suggest that rubber composites in which 10–20 phr of CB are replaced by HC might be promising materials. The use of HC in the rubber industry would represent a high-tonnage application for hardwood waste.

## 1. Introduction

In view of recent developments in the legal and economic situation, the rubber industry currently seeks new, naturally grown raw materials. Exploiting production waste, such as hardwood sawdust used in this work, in the role of a precursor of the filler phase for natural rubber, would be of significant economic advantage.

Filler phases play a crucial role in technical rubbers, such as the ones used for tire production. They markedly improve their physical and mechanical properties. Because of the high weight percentage of the filler phase in a typical technical rubber (in tires: ca. 30%), the filler significantly affects the economy of rubber production. The basic properties that define fillers are size, structure, surface activity, specific surface area, and particle shape. Tearing of polymer chains during bending and stretching can be caused by the filler if its particle size is significantly larger than the distance between the ends of elastic chains. This leads to a stressed surface and cracking. Carbon black is a traditionally used filler in the rubber industry due to its remarkable reinforcement properties, carbon purity, and low cost [1]. CB is usually obtained through thermal oxidative processes, including partial combustion of liquid aromatic hydrocarbon oil in the oil furnace process in the temperature range from 1400 to 2000 °C [2].

In the last few decades, great efforts have been made to replace carbon black with bio-sourced fillers which ideally should be processed at less energetically demanding conditions. Such bio-fillers include bamboo fiber [3], soy protein and carbohydrate [4,5], pineapple leaf fiber, waste eggshell powder [6], starch [7,8], and cereal straw [9]. Another material from this class, biochar, can be produced by pyrolysis of all biomass types in the presence of a small amount of oxygen or in its complete absence since this process converts any organic matter into a highly stable carbon form. Based on data from the detailed literature review, biochar can be obtained from different feedstocks, such as dry leaves [10], coconut shells and discarded wood pallets [11], birchwood [12], corn stover [13], rice husks and bran [14,15], lignin sources [16], woody waste biomass [17], or coppiced wood [18,19]. Some studies have shown that hardwood as feedstock contains higher carbon and lower ash content in comparison to other low-value biomass sources, which leads to improved reinforcing properties [18]. The primary challenge in using biochar as a substitute for the CB filler in rubber composites is the optimization of its particle size since biochar is obtained from pyrolyzed biomass which often was coarse-grained. Therefore, the particle size of biochar is reduced by milling or grinding, which can be costly if the typical CB particle size (10–100 nm) has to be matched. Milled biochar typically contains a small fraction of particles in the size range from 10 to 100 µm, which might cause localized stresses in the rubber composites and eventually lead to product rupture [17].

Recently, considerable research attention turns to rubber reinforcing phases based on carbonaceous materials obtained through milder thermochemical conversions, such as the hydrothermal carbonization (HTC) [20] of different biomass sources, such as lignin isolated from wood [21], cellulose, hemicellulose, or soybean protein [22]. The HTC process provides an attractive opportunity to carbonize any type of biomass, even those with high water content at distinctly lower temperatures (180–300 °C) than the ones required for classical pyrolysis. HTC consists of a series of reactions which decrease the hydrogen and oxygen content of the feedstock, namely hydrolysis, aromatization, decarboxylation, dehydration, and re-condensation [23]. Water, which acts as a reaction medium, additionally generates a considerable autogenous pressure. The final, lignite-like product obtained by HTC is called “hydrochar”, and was found to resemble the carbon black (CB) filler in many properties [23,24,25,26]. Great advantages of the HTC process are that the medium, water, is cheap and non-toxic, and also that the usually wet raw biomass does not need to be dried before this process, which would be energy-consuming [27]. The main process parameters of HTC are the temperature and the residence time, which affect the degree of coalification of the raw biomass (i.e., its final carbon content). Generally, hydrothermal carbonization is considered to be an environmentally friendly and cheap process, which is easy to be applied [28].

The aim of the authors in the present work is to investigate the possibility of partially replacing CB with bio-sourced hydrochar (HC) which was obtained by the hydrothermal carbonization of hardwood sawdust waste. The latter presents a cheap filler precursor with a relatively high initial carbon content. For fulfilling the mentioned aim, a brief hydrothermal carbonization treatment was applied to the sawdust, followed by ball milling. The obtained hydrochar was then mixed with natural rubber and with additives, following a commercial recipe. The obtained composites also contained CB as the second filler component. The weight loading of the combined fillers in the rubber composites was kept constant, while the ratio CB/hydrochar was varied, in order to evaluate to which extent CB can be replaced by hydrochar without detriment to material properties. The authors’ present work is divided into two parts. In this actual paper (first part of the work), a comprehensive characterization of the prepared HC filler is undertaken, as well as the analysis of morphology and vulcanization behavior of the rubber composites. A second follow-on paper is dedicated to the mechanical and thermo-mechanical properties of the composites, their swelling behavior, as well as their chemical stability. In this paper, the HC particle size and geometry were assessed by TEM, its specific surface area was determined (BET), and the elemental composition of HC was analyzed by combustion, while its chemical structure was studied by means of ^13^C-NMR and FTIR spectroscopy. The morphology of the prepared rubber nanocomposite samples (i.e., the dispersion of both fillers at different CB/HC ratios) was assessed by means of SEM and TEM. The influence of different CB/HC ratios on the vulcanization behavior was elucidated by means of rheology tests. The summarized results suggested that the CB filler can be replaced with hydrochar (HC) up to CB/HC ratios of 40/10 or 30/20 (10–20 phr of HC) without significant detriment to product properties.

## 2. Materials and Methods

### 2.1. Materials

Natural rubber Standard Vietnamese Rubber CV60, (Vietnam Rubber Group, Ho Chi Minh City, Vietnam), as well as the following additives: N-isopropyl-N′-phenyl-p-phenylenediamine (IPPD), stearic acid, zinc oxide (ZnO), sulfur, and N-cyclohexylbenzothiazol-2-sulfenamide (CBS), were all kindly obtained from Edos (Zrenjanin, Republic of Serbia), and used as received without further purification. Carbon black N330, the conventional filler component, with a typical particle size of 30–50 nm, was purchased from Nhumo (Altamira, Mexico). The preparation of hydrochar, used as a potential replacement for CB, is described in detail in the next section.

### 2.2. Preparation of the Hydrochar Filler

In order to obtain the hydrochar filler, hydrothermal carbonization treatment of hardwood waste biomass (sawdust originated from the oak tree, cellulose content: 91 wt.%, lignin content: 9 wt.%) was performed at 300 °C, under autogenous pressure of 86.6 bar, with a process duration of 30 min. The obtained hydrochar consisted of relatively large particles and their agglomerates. In the next step, the raw product was ground in a planetary mill (model Mono Mill Pulverisette 6), from Fritch (Idar-Oberstein, Germany), at 200 rpm for 5 min. The grounded particles were then sieved from 500 to 800 µm and thereafter they were washed with hot deionized water until the dark leachate (containing degraded lignin and cellulose fragments) was fully extracted. Finally, the product was dried in an oven for 24 h. The total hardwood weight loss after HTC treatment and sieving was 62.3%, which means that the hydrochar yield was 43.8%.

### 2.3. Composition of the Rubber Mixtures

The recipe for obtaining the studied natural rubber composite samples is presented in Table 1, namely the phr amounts of all constituents and additives.

As presented in Table 1, the additives used in this work are N-isopropyl-N′-phenyl-p-phenylenediamine (IPPD), stearic acid, zinc oxide (ZnO), sulfur, and N-cyclohexylbenzothiazol-2-sulfenamide (CBS). A necessary component in any technical rubber is the reinforcing filler. In this work, two types of rubber fillers are used: hydrochar and carbon black. The total filler amount was kept at 50 phr (according to ASTM-D1765), while their ratio was varied (10, 20, 30, 40, and 50 phr of CB plus 40, 30, 20, and 10 phr of HC, respectively). The content of the remaining additives was kept constant. In preliminary tests, the composition filled exclusively with 50 phr of HC systematically displayed distinctly poor end-use properties, especially mechanical ones, and hence it was discarded from the series of specimens, which were investigated in detail in this work.

The prepared rubber samples were labeled according to the carbon black content (classical filler in the filler mixture) and their vulcanized/unvulcanized state, as shown in Table 2. The unvulcanized samples contained the same amounts of all additives as the samples destined for vulcanization, but they were not subjected to the vulcanization process at 150 °C.

**Table 2 polymers-15-01176-t002:** Labeling of the prepared rubber samples and the content of co-fillers in them expressed in phr, wt.%, and % *v*/*v*.

Sample Code *	CB Content, phr	HC Content, phr	CB Content, wt.%	HC Content, wt.%	CB Content, % *v*/*v* ^‡^	HC Content, % *v*/*v* ^‡^	Combined Fillers % *v*/*v* ^‡^
CB10 * and VCB10 *	10	40	6.29	25.16	2.42	20.39	22.80
CB20 and VCB20	20	30	12.58	18.87	4.83	15.29	20.12
CB30 and VCB30	30	20	18.87	12.58	7.25	10.19	17.44
CB40 and VCB40	40	10	25.16	6.29	9.66	5.10	14.76
CB50 and VCB50	50	0	31.45	0	12.08	0	12.08

* CBxx = unvulcanized samples, VCBxx = vulcanized samples. **^‡^** Calculated using 1.14 g/mL as the density of all the composites and the experimentally determined (see Table 3) densities of CB (2.969 g/mL) and HC (1.407 g/mL).

**Table 3 polymers-15-01176-t003:** Results of S_BET_ determination.

Sample	Density, g/cm^3^ *	S_BET_, m^2^/g
Carbon black	2.969	77.8
Hydrochar	1.407	21.4

* determined by pycnometry in nitrogen gas.

### 2.4. Mixing and Vulcanization Procedure

The mixing procedure included three steps. (1) Mixer preparation: idle run, conditioning at 90 °C, (2) component preparation: mixing neat natural rubber at several speeds until constant process parameters, and (3) component mixing: admixing the remaining components in Table 1. The mixing procedure was performed by the Laboratory mixer Haake Rheomix (model 600, Thermo Fisher Scientific, Waltham, MA, USA), modified with a drive unit Haake Rheocord EU-5. The employed three-step mixing procedure was identical to the one described in detail in our previous published work [29].

Vulcanization: The samples of the rubber composites were vulcanized, following the ISO 37 standard, by pressing the rubber sheets for 15 min at 150 °C, at atmosphere pressure, cut into dumbbells, and left for 24 h at room temperature. Subsequently, the vulcanized sheets were left to relax for 24 h at room temperature. Thereafter, the samples were cut into specimens of desired shapes and sizes. Unvulcanized rubber composites were also studied in this work as reference materials.

### 2.5. Rheology of Vulcanization

The rotorless rheometer MDR-A, supplied by Beijing Rade Instrument Co. Ltd., Beijing, China, was used for monitoring the vulcanization process. The rheological tests were performed during the first 15 min of the process, at 150 °C, for all prepared samples.

### 2.6. Structure and Morphology of Fillers and Nanocomposites

13C-NMR: Solid-state NMR (ssNMR) spectra were collected using a 700 MHz Bruker Avance Neo NMR spectrometer (Bruker, Karlsruhe, Germany) (B_0_ = 16.4 T) at Larmor frequency ν (^13^C) = 224.684 MHz, using a double-resonance 3.2 mm magic angle spinning (MAS) probe. All MAS NMR experiments were recorded with the SPINAL 64 decoupling sequence. All ^13^C ssNMR experiments were performed at 17 kHz and 4 s recycle delay. ^13^C CP/MAS NMR spectra were recorded with 1.5 ms spin-lock at 4096 scans. The ^13^C chemical shift was calibrated using α-glycine (176.03 ppm; carbonyl signal) as an external standard. The samples were kept and packed into ZrO_2_ rotors under an Ar atmosphere. The NMR experiments were performed at a temperature of 303 K and temperature calibration was performed to compensate for the frictional heating of the samples [30]. All the NMR spectra were processed using the Top Spin version 3.5 pl7 software package.

FTIR: The infrared (IR) spectroscopic characterization of natural rubber composites was performed in the ATR FTIR mode (reflectance), using a Nicolet 6700 spectrometer (Thermo Scientific, Madison, WI, USA, now Thermo Fisher Scientific, Waltham, MA, USA) in a dry air-purged environment equipped with reflective ATR extension GladiATR (PIKE Technologies, Fitchburg, MA, USA) with a diamond crystal. Spectra were recorded in the 4000–400 cm^–1^ region with a DLaTGS (deuterated L-alanine-doped triglycine sulfate) detector at resolution 4 cm^–1^, 64 scans, and Happ–Genzel apodization.

Powder samples of carbon black and hydrochar were characterized on the same spectrometer in the transmission mode (FTIR) and prepared as KBr pellets (0.2% of sample material in KBr).

The real density was determined by an automatic pycnometer (Pycnomatic ATC, from Thermo Fisher Scientific, Waltham, MA, USA) at (25 ± 2) °C, using N_2_ as a test gas.

Analysis of specific surface area (BET): Prior to the characterization of internal pore structure, both samples were vacuum-dried at 100 °C for 10 h. The specific surface area (SBET) was measured by a gas adsorption technique on a Gemini VII 2390 (Micromeritics Instruments Corp., Norcross, GA, USA) with nitrogen as the sorbate. The surface area was calculated from the Brunauer-Emmett-Teller (BET) (Quantachrome, CA, USA) adsorption/desorption isotherm using Gemini software. It characterizes materials in the region of micropores (<2 nm) [31].

Filler size, shape, and dispersion in rubber:

TEM: The morphology of nanoparticles and nanocomposites was studied using the transmission electron microscope Tecnai G2 Spirit Twin 120 kV (FEI Czech Republic, Brno, Czech Republic) operated at an acceleration voltage of 120 kV.

Nanofiller sample preparation for TEM: The powdery nanofillers were dispersed in ethanol and deposed as a drop onto a TEM grid (400 mesh) consisting of a copper grid covered with a thin, electron-transparent carbon film. The drop containing nanofiller was dried at room temperature before TEM observation of the so-deposed filler particles.

The bulk samples (CB40, VCB40, CB10, and VCB10) were examined in two ways.

Rubber nanocomposite specimens’ preparation for TEM: The nanocomposites were cut into ultrathin sections (specimens) using an ultramicrotome (Ultracut UCT, from Leica, Wetzlar, Germany) under cryo-conditions (the sample and knife temperatures were −80 and −50 °C, respectively). The ultrathin sections were transferred onto a microscopic grid covered with a thin carbon film in order to improve their stability under the electron beam during TEM observations.

SEM: Fracture surfaces of the studied nanocomposites were examined using the scanning electron microscope MAIA3 from Tescan (Brno, Czech Republic) in the high-vacuum mode at an operating voltage of 3 KeV.

Rubber nanocomposite specimens’ preparation for SEM: Prior to observation, bulk samples were broken in liquid nitrogen (brittle fracture) and a conductive thin Pt film was deposited on their surface using the vacuum sputter coater SCD 050 (from Leica, Wetzlar, Germany).

EDX elemental analysis in the SEM microscope: The SEM samples were also examined with energy-dispersive X-ray analysis of (EDX; detector X-MaxN 20; from Oxford Instruments, Abingdon, Oxford, UK).

Determination of C, H, and N content (combustion analysis): The FlashSmart™ Elemental Analyzer (Thermo Fisher Scientific, Waltham, MA, USA) was used for the concurrent determination of C, H, and N content. Approximately 1.5 mg of the sample was weighed in a tin capsule using the precision weighing balance Sartorius SE 2-OCE (from Sartorius AG, Göttingen, Germany). The capsule was sealed and inserted into the autosampler of the Analyzer and the standard operating procedures were employed. The elemental analysis was performed twice for each sample.

Ash determination: Approximately 100 mg of the sample was accurately weighed into a ceramic crucible with a Mettler Toledo-H20 analytical balance (from Mettler Toledo, ZH, Greifensee, Switzerland) and closed with a ceramic lid. The crucible was heated with a Meker burner for 15 min (in an air atmosphere) and then allowed to cool in a desiccator for at least 30 min. This procedure was repeated until the constant weight of the crucible was reached. Ash determination was performed twice for each sample.

## 3. Results and Discussion

### 3.1. Hydrochar Filler

This work is dedicated to rubber composites filled with a new-generation hydrochar filler (HC), which was used as a biomaterial-based partial replacement of the traditional carbon black (CB) nanofiller (see Figure 1). The dark brown hydrochar was prepared via 30 min hydrothermal carbonization of sawdust from hardwood (oak tree, cellulose content: 91 wt.%, lignin content: 9 wt.%, as determined below via elemental analysis), which was performed at 300 °C under the pressure of 86.6 bar. An important task was to characterize the basic properties of the new filler and to compare them with the commercial carbon black.

#### 3.1.1. Size and Geometry of the Hydrochar Particles

The shape and geometry of the neat fillers, hydrochar, and carbon black, are compared in Figure 1. For the observation, the powdery neat fillers were dispersed in ethanol (3 min ultrasonication) and subsequently deposed in a drop on a microscopic copper grid covered with a carbon film, after which ethanol was evaporated (thus leading to some agglomeration). Distinct differences in the filler particle size and shape can be seen: HC has much larger (0.5–3 µm) and less regular particles than CB (30–60 nm). This difference suggests that the specific surface per gram of HC should be 15 to 105 times smaller (depending on size distribution; estimation: see Appendix A) than in the case of carbon black. The reduction in the filler–polymer interface area is an important and potentially detrimental aspect concerning the mechanical effect of the filler (mechanical tests: see follow-on work [32]). The specific surface area of both fillers hence also was directly determined and compared (see below).

#### 3.1.2. Specific Surface Area of fillers

The specific surface area of a filler is very important for its physical and chemical crosslinking with the matrix. This magnitude was determined for the commercial carbon black (CB) used in this work, as well as for the hydrochar (HC) prepared by the authors. The values of the specific surface area were determined from nitrogen adsorption isotherms at T = 77 K using the Brunauer–Emmett–Teller (BET) theory (as S_BET_, see Table 3).

The S_BET_ values in tens of m^2^/g, as well as the shapes of the isotherms (see Appendix A) indicate that the materials belong to the meso- or macroporous solids, which are subject to multilayer adsorption [33]. HC expectedly displays a lower specific surface than CB, but the difference between HC and CB (factor 3.6) is much smaller than would be expected from the above-discussed particle size (factor 15 to 105). This result hence indicates that HC is significantly porous. In view of the particle sizes observed in Figure 1, the theoretical specific surface area of HC was calculated (using also the HC density from Table 2) to be 1.42–8.53 m^2^/g, depending on average particle size, for the idealized case of regular non-porous cubic particles (see Appendix A). In the same way, the theoretical specific surface area of CB was estimated to be 33.7–67.4 m^2^/g, depending on average particle size, which is in much better agreement with the BET analysis. The mentioned theoretical values correspond to the external surface area of the filler particles, which is important for their reinforcement capability in a high molecular weight polymer matrix (as the internal surface of the pores is not easily accessible for interactions with the polymer). Hence, in spite of the comparable values of S_BET_, HC is expected to possess a significantly smaller reinforcement capability.

Noteworthy is the difference in the densities of both co-fillers (see Table 3): CB is two times denser than HC. This means that at high HC contents, the volume fraction of the fillers will be markedly larger than in the case of a composite filled exclusively with CB. The calculated volume fractions of the fillers are listed above in Section 2.3 in Table 2. If comparing samples CB50/VCB50 and CB10/VCB10, it can be seen that the volume fraction of the filler nearly doubles if going from the former (50 phr of CB) to the latter (10 phr of CB + 40 phr of HC), from 12.1 % *v*/*v* (CB50/VCB50) to 22.8 % *v*/*v* (CB10/VCB10). The larger volume of the HC filler in CB10/VCB10 cannot be expected to compensate (via hydrodynamic effect) for the much smaller specific external surface area of the HC particles.

#### 3.1.3. Chemical Composition of the Hydrochar

##### Elemental Composition of the Fillers

An important characteristic of the partly carbonized hydrochar filler, as well as of its degree of carbonization, is the elemental composition. A combustion analysis was performed, which sensitively determined the contents of carbon, hydrogen, and nitrogen (CHN analysis, see results in Table 4). Additionally, the analysis of ash content was also carried out in order to determine mineral content (see last column results in Table 4). Finally, the content of oxygen was roughly estimated as the missing wt.% of combustible material (see Table 4).

The results in Table 4 indicate a marked difference between the (partly carbonized) hydrochar synthesized by the authors and the commercial carbon black. The hydrochar was found to be organic-like, in contrast to carbon black, which was graphitic (and also conductive). The data in Table 4 further indicate that the hydrochar went approximately 50% of its transformation path from wood to carbon black if the contents of carbon and oxygen are concerned. The hydrogen content, which is essential for the ‘organic character’ of hydrochar, decreased only moderately (by 1/6 part) if going from sawdust to hydrochar.

The composition of the dark brown hydrochar corresponds (using data from Table 4 and atomic masses) to the empirical formula C_12_H_10_O_3_ (N_0.02_ within the error margin), which is not far from slightly hydrogen-deficient aromatic compounds. The nitrogen content in parentheses is very close to the sensitivity threshold. In contrast, the composition of commercial carbon black corresponds to the (12-C-atoms-) formula C_12_ (H_0.78_ within error margin) (O_0.33_ within error margin); however, the elements in parentheses are determined beyond the sensitivity range, so that CB, in reality, could be practically pure carbon.

The elemental composition of the studied hydrochar can be compared with the ones of the constituents of the intact hardwood sawdust. Cellulose has the composition C (44.5%), H (6.2%), O (49.3%), the repeat unit C_6_H_10_O_5_, and the ‘12-C-formula’: C_12_H_20_O_10_. Cellulose contains much less carbon and much more oxygen than the studied hydrochar. Hypothetical ‘average lignin’, based on coniferyl alcohol which lost three hydrogens would have the composition C (68%), H (5%), and O (27%), its repeat unit would be C_10_H_9_O_3_, and the ‘12-C-formula’: C_12_H_10.8_O_3.6_. The markedly different elemental composition of cellulose and lignin also made possible the determination of their approximate ratio in the sawdust prior to its hydrothermal carbonization: 91 wt.% cellulose + 9 wt.%. Lignin (corresponding to 91.7 + 8.3 mol%) is the best mathematical fit. The ‘averaged lignin’ material comes close to the composition found for the studied hydrochar, but the initial content of lignin in sawdust was low, as just mentioned, and additionally, lignin is known to relatively easily undergo pyrolysis (including hydrothermal one) at 200 to 400 °C [34], so that it would be expected to be mostly quantitatively degraded during the production of the studied hydrochar. Hence the hydrochar’s chemical structure with nearly doubled carbon content and nearly halved oxygen content must originate practically exclusively from ‘caramelized’ cellulose. The ^13^C-NMR and IR spectroscopy (discussed below) were used to further elucidate the chemical nature of the prepared hydrochar. In the literature dedicated to hydrochar production, it can be found that the carbon content in hydrochar obtained from hardwood waste HTC treatment can be up to 87% [24], depending on the harshness (and energy input) of the process. As the cited HTC processes took place at a distinctly longer time (2 h) than the relatively quick hydrothermal carbonization employed in this work (30 min), the achieved carbon content can be considered to be an interesting result.

##### Structure Characteristics

^13^C-NMR: The ^13^C CP/MAS NMR spectroscopy was used to investigate the differences in the chemical structure of raw and hydrothermally carbonized hardwood sawdust. The ^13^C CP/MAS NMR spectra of both materials are depicted in Figure 2. (It was not possible to characterize the commercial carbon black filler with the conventional experimental setup due to the conductive and diamagnetic character of CB. In the literature, milled graphite is reported to display a very broad unsymmetrical peak ranging from −10 to 200 ppm, with a maximum near 160 ppm [35].) In Figure 2, it generally can be seen that the wood material underwent dramatic chemical change during the hydrothermal carbonization, but did not become graphitic, and the resulting structure is random and disordered.

The spectrum of raw sawdust from an oak tree (Figure 2a) confirms that cellulose (assignment, see, e.g., [36]) is the dominant fraction in this starting material (the above elemental analysis yielded the cellulose content of 91 wt.%), while the fraction of lignin (assignment, see, e.g., [37]) is only a minor one (the above elemental analysis: 9 wt.% of lignin). The spectrum is well-structured with relatively narrow peaks, which is characteristic of the presence of few well-defined chemical compounds.

The hydrothermally carbonized material (hydrochar: Figure 2b) displays a ^13^C CP/MAS NMR spectrum which is highly different from the intact sawdust. Three product peaks, two of which were broad, intense, and structured, as well as a small one, were detected. The first broad signal, which ranges from 0 ppm to 65 ppm, can be assigned to various types of aliphatic carbon atoms. The second broad signal extends from 95 ppm to 165 ppm and can be assigned to various types of unsaturated carbon atoms (such as aromatic or olefinic). The broad signals suggest the presence of polymeric and/or widely structurally varied organic species. Finally, the mentioned small peak at 176 ppm corresponds to carbonyl groups. In the spectrum of sawdust, a weaker and narrower carbonyl signal centered around 172 ppm also is found. The prominence of carbonyl groups was evaluated in more detail by FTIR (see below). Noteworthy are two relatively sharp signals in the 13C CP/MAS NMR spectrum of hydrochar (Figure 2b), which protrude from both broad signals, at 55 ppm and 145 ppm, respectively. These signals suggest the formation of well-defined structural units with similar/same environments. The signal at 55 ppm can be attributed (in the context of the determined elemental composition) to alkyloxy alpha carbon atoms joined to unsaturated units, such as phenyl or C=C. Only a minor (and slightly shifted) peak is found near this position in the spectrum of sawdust. The second relatively sharp signal at 145 ppm can be assigned to carbon atoms in frequently occurring specific aromatic or olefinic structure units. This signal does not correspond (precise position, as well as intensity) to any of the lignin peaks.

Transformation of the wood components: While some persisting weak signals of lignin (its initial content was relatively low, as demonstrated above by elemental analysis) could be covered by the two broad peaks observed for the hydrochar, there are several major peaks attributed to cellulose (from 65 to 88 ppm, especially intense at 73 and 75 ppm) which are completely missing in the hydrochar spectrum. Hence it can be concluded that cellulose was completely transformed into a much carbon-richer polymer, as also indicated by the (above-discussed) elemental composition, while lignin (initially making up only 9 wt.%), in view of its instability at the conditions of the hydrothermal treatment (see [34]), either should have escaped via depolymerization or its molecular fragments were also incorporated into the hydrochar structure (as in [21]). In view of the dominant fraction of cellulose, as demonstrated by the spectrum in Figure 2a and by the elemental (CHN) analysis, the new structures observed in the hydrochar (Figure 2b) must have originated nearly exclusively from the cellulose.

Degree of carbonization: If compared with the literature spectra [38] of disordered carbon materials produced by pyrolysis, the ^13^C CP/MAS NMR spectrum of hydrochar is distant from pyrocarbon but close to typical char (more organic), in contrast to the studied hydrochar having a more structured spectrum and being somewhat richer in aliphatics.

FTIR: The differences in chemical structure, as well as the extent of carbonization in hydrochar were also investigated by means of IR spectroscopy. The hydrochar and carbon black fillers were characterized in the reflectance mode as KBr pellets, while sawdust from hardwood (precursor of the hydrochar) was measured in the transmittance mode as KBr pellets. The FTIR spectra of the mentioned materials (zoomed to similar sizes in reflectance mode, or, in the case of sawdust, recalculated to absorbance, which is approximately equivalent) are compared in Figure 3.

The IR spectra in Figure 3 confirm the vast difference between the graphitic carbon black (CB) on one hand, and the organic hydrochar and sawdust, as well as marked differences between the two latter materials.

As expected for graphite, CB displays only one weak peak at 1630 cm^−1^, (C=C) and no additional significant absorptions. The spectra of wood (from an oak tree: 91 wt.% of cellulose + 9 wt.% of lignin, as determined above by elemental analysis) and of hydrochar display marked differences in the fingerprint region, where both have most peaks between 1800 and 1000 cm^−1^. In case of wood (sawdust), there is only one dominant peak around 1031 cm^−1^ (C–OH, some types of C–O–C stretching), while the other numerous peaks are relatively small (and they also have somewhat different positions than the fingerprint peaks in hydrochar). In the fingerprint region of hydrochar, on the other hand, the peaks are most intense in the region 1800–1100 cm^−1^ (twisting, C–O–C stretching, C=C, stretching, eventually carbonyl), where the peaks of sawdust are weak. The vast spectral difference suggests strongly different organic structures in wood and hydrochar. An interesting feature is the hydrochar peaks at 1696 and 1604 cm^−1^. They could be assigned to conjugated carbonyl and to C=C stretching, respectively. The peak at 1696 cm^−1^ is not very intense compared a carbonyl peak, so only a moderate content of pyrolysis-generated carbonyl groups (seen also by ^13^C-NMR) can be postulated.

### 3.2. Nanocomposite Rubber/CB/Hydrochar: Basic Properties

In the rubber composites prepared in this work, the amount of the combined fillers CB and HC was kept constant at 50 phr (31.45 wt.%), while their ratio was varied (10, 20, 30, 40, and 50 phr of CB plus 40, 30, 20, and 10 phr of HC, respectively). The composites filled exclusively with 50 phr of HC (which would be coded “CB00”/“VCB00”) systematically displayed distinctly poor end-use properties in preliminary tests, especially mechanical ones. Hence, CB00 and VCB00 were discarded from the series of specimens investigated in detail in this work. It seems that at least a small content of CB is needed to achieve proper vulcanization under the standard preparation conditions used in this work.

#### 3.2.1. Dispersion of the Fillers and Overall Morphology

The dispersion of the combined fillers in the prepared rubber composites was analyzed by means of electron microscopy. Figure 4 shows the typical morphology in the transmission mode (TEM) on the example of the sample with 40 phr of carbon black (combined with 10 phr of hydrochar) in the unvulcanized and vulcanized state: samples CB40 and VCB40, respectively. Similar TEM images of a sample rich in hydrochar, 10CB/10VCB (10 phr of carbon black + 10 phr of hydrochar), are shown in the Appendix A. In Figure 5, the morphology of a fracture surface of 40CB/40VCB is imaged in the SEM mode, while Appendix A shows similar SEM morphology images of hydrochar-rich 10CB/10VCB. Generally, the results show that both carbon-based nanofillers are reasonably well visible by TEM in the organic matrix (in 60 nm thick specimens) and that their dispersion was even. SEM also makes possible the visualization of the relatively large hydrochar grains, as well as their assignment by EDS elemental analysis.

The exemplary TEM images of CB40/VCB40 in Figure 4 make it possible to identify the particles of both carbon-based fillers in view of their size and geometry, which was determined for the neat fillers (see above). It can be noted that in spite of the smaller size, carbon black (CB) displays a sharper contrast in the organic matrix, obviously due to its conductivity as well as due to its higher density. The irregular hydrochar particles nevertheless also are visible and easily recognized. The morphology analysis indicates an even distribution of both carbon black, as well as hydrochar. In the unvulcanized state, occasional multi-micrometer-sized bubbles represent a typical morphology feature, while any bubbles are practically absent after vulcanization. Occasional intensely dark particles in Figure 4 can be assigned to zinc oxide (ZnO) in the unvulcanized and vulcanized state (see Figure 4a,b,d), while in the unvulcanized state, some of them (less dark ones) could be assigned to sulfur micro-crystallites. Appendix A shows a similar morphology of the hydrochar-rich sample VCB10. Some differences can be seen in comparison to VCB40. While hydrochar is evenly distributed in VCB10, the carbon black is separated into CB-rich domains, which are separated by a continuous CB-deficient matrix.

Figure 5 shows SEM images of the fracture surface of the prepared nanocomposites on the example of CB40/VCB40. The scale is larger in Figure 5 than in Figure 4. It reaches from a 1 mm scale bar down to a 20 µm scale. It can be observed that the largest bubbles in the unvulcanized samples reach a size of 100 µm (somewhat wider than the thickness of a typical human hair). Upon careful observation, the embedded hydrochar particles can be recognized in the fracture surface at higher magnification (Figure 5b,c,e,f). They are better visible in the smoother, bubble-free fracture surfaces of the vulcanized samples. Appendix A shows a similar morphology of the hydrochar-rich sample VCB10. Some differences can be seen in comparison to VCB40. Hydrochar is evenly distributed, but some occasional HC agglomerates can be seen. The hydrochar filler also is much more easily discerned in the fracture surfaces of vulcanized and non-vulcanized samples. The identity of the hydrochar particles was confirmed by EDX elemental analysis (see Appendix A), which indicated a high oxygen and carbon content in the particles, while practically no oxygen was found in the surrounding neat matrix.

#### 3.2.2. Vulcanization Behavior

The effect of the added nanofillers, and specifically of the hydrochar, on the viscoelastic and curing properties was evaluated by means of vulcanization tests in a moving die rheometer. The experiments were carried out at 150 °C and their duration was 15 min, like in the case of the standard cure procedure employed for preparing the vulcanized samples in this study. The measured dependence of rheological torque on time is compared in Figure 6 for all the tested compositions. The results suggest that both a rheological (reduced filler surface due to larger hydrochar grains), as well as a chemical effect (shortened induction time = ‘scorch time’ after process start), occur.

All samples display the expected increase in the rheological torque with time, which is characteristic of vulcanization (Figure 6). Higher overall and final torque values are registered for the samples containing higher carbon black (CB) loading. The sample VCB10 with 10 phr of CB (and 40 phr of HC) displayed distinctly slower vulcanization, as well as markedly lower initial and final torque values than the CB-richer samples. This is can be assigned to the lower external surface of the combined fillers caused by the larger hydrochar grains, which leads to their reduced reinforcing capability, physical (via adsorption) in the initial stage and chemical in the later stages (via sulfur bridges polymer filler). An effect of the increasing volume fraction of the combined fillers in the HC-rich composites is not unambiguously recognizable in the vulcanization curves, but it might contribute to the finding that the initial viscosity only slightly drops if going from VCB30 to VCB10.

Both the initial and final torque values markedly increase in VCB40 (in comparison to VCB10), which already displays a similar behavior, such as VCB50. Nevertheless, the final torque of VCB40 still is significantly lower due to the smaller specific surface area available for chemical crosslinking. A purely physical (rheological) effect of the carbon black nanofiller, which has a very high specific surface available for adsorption, is visible in the region of the process start. VCB40 and VCB50 display significantly higher initial torques than the other samples, whose content in carbon black is smaller. The vulcanization curve of the samples VCB40 and VCB50 is steep and S-shaped and both samples also display a distinct and similar scorch time before the torque increase starts. In view of the vulcanization behavior, the sample VCB40 appears to be a highly promising product.

VCB20 displays a markedly higher final torque value than VCB10, but is still distinctly smaller than VCB40. Additionally, the starting torque value of VCB20 value is similarly small, like in the case of VCB10.

The vulcanization curve of VCB30 (30 wt.% hydrochar + 20 wt.% carbon black) displays an increase in viscosity which is significantly higher than in the case of VCB20, but still considerably smaller than in the case of VCB40. Interestingly, the vulcanization of VCB30 starts practically without an initial delay (scorch time is close to zero). Nevertheless, the torque generated by VCB30 never surpasses the torque displayed by VCB40 or VCB50 at any point in time because of the lower overall resistance (torque) of VCB30. However, the product properties of VCB30 are not constant at any moment, so the desirable safety scorch time is missing. The behavior of VCB30 suggests that hydrochar has an increased reactivity (e.g., due to non-aromatic C=C bonds, see structure discussion above). This reactivity might theoretically even lead to premature scorch during the mixing of the rubber recipe, which, however, was performed at a temperature of just 90 °C. The collected data are not sufficient to prove or disprove eventual premature scorching in VCB30 during recipe mixing. The shortened scorch time is less visible but still discernable in the case of the samples VCB20 and VCB10, whose vulcanization curves generally are flat.

While the cure behavior of VCB40 (with 10 phr of hydrochar) is very promising, the HC-rich products would need further optimization of the vulcanization chemistry. The drastically reduced scorch safety time in VCB30 might be critical during its processing. Changes in the recipe (vulcanization additives) or additional treatment of the filler would be necessary to prevent premature scorching and to improve the final degree of crosslinking.

The rheological parameters obtained for the vulcanization of the prepared natural rubber composites are summarized in Table 5. The values of the cure rate index (CRI, ‘vulcanization speed’) decrease with higher hydrochar content in the samples, similar to maximum and minimum torque. This seems to correlate with the reduced specific surface of the filler and also with its postulated reactivity. This reactivity causes the vulcanization reactions to start immediately, but it also can consume a part of the vulcanization agents. For the same reasons, an opposite hydrochar influence is noticed for the cure time values, which increase with hydrochar content. Eventually, the chosen process time of 15 min might be not sufficient to reach the maximum theoretically possible crosslinking in the samples with the highest hydrochar content, such as VCB10. The described difference in rheological parameters affects the vulcanization process (Figure 6) and also suggests some trends in the mechanical properties of the final natural rubber products (see follow-on work [32]).

#### 3.2.3. FTIR Analysis of the Nanocomposites

The prepared rubber nanocomposites, prior to and after vulcanization, were characterized by infrared spectroscopy (FTIR, see Appendix A). Unvulcanized and vulcanized materials with the same contents of hydrochar and carbon black (CB) display practically identical FTIR spectra. A general trend is that with increasing CB fraction, the intensity of all peaks decreases due to the reflecting properties of CB and due to its lack of intense FTIR peaks.

The FTIR spectra correspond to a simple superposition of rubber and hydrochar peaks; no additional ones appear as a result of filler-matrix interaction or vulcanization. In addition to the above-discussed FTIR absorption ranges of hydrochar, the following specific peaks of natural rubber appear: two intense peaks at 1435 and 1374 cm^−1^ (twisting of CH_2_ and CH_3_ groups, respectively), smaller peaks at 1079–997 cm^−1^ (skeleton twisting), and an intense peak at 834 cm^−1^, which much exceeds the small peak of hydrochar at this place (and the ‘tail’ of this peak at 752–660 cm^−1^ (all skeleton twisting). The C–H stretching peak at 3030–2800 cm^−1^ is much more intense and structured than in the case of hydrochar due to numerous C–H bonds in natural rubber.

## 4. Conclusions

Rubber composites were prepared and studied, in which the traditional carbon black (CB) filler was partly replaced by bio-sourced ‘hydrochar’ (HC), which in turn was produced from hardwood waste (from an oak tree, cellulose content: 91 wt.%, lignin content: 9 wt.%) by the innovative and energy-saving hydrothermal carbonization route (HTC). In this part of a broader work, a comprehensive characterization of the prepared HC filler is undertaken at first, as well as the analysis of rubber composites’ morphology and their vulcanization behavior, at different fractions of CB/HC ratios in the filler phase. Other interesting characteristics of the rubber composites are studied in a follow-on work.The particle size and shape of HC and CB were compared using TEM. The HC grains were found to be larger (0.5–3 µm) and less regular than the CB particles (30–60 nm).The specific surface area of CB (determined by the BET method) was expectedly found to be larger than that of HC (77.8 vs. 21.4 m^2^/g), but much less so than suggested by the difference in particle size. This finding indicates a considerable porosity of HC.Elemental analysis of the bio-waste feed (sawdust), the hydrochar produced from it, and the traditional CB filler revealed that the HTC treatment substantially raised the carbon content in the final hydrochar (HC) product, from 46 to 71 wt.%, but HC still stayed relatively organic in its composition if compared to CB (more than 96 wt.% of C). The oxygen content was halved after HTC processing (from 47 to 24 wt.%), but remained significant.Analyses of a primary structure performed by means of FTIR and ^13^C-NMR confirmed the predominantly organic character of HC, but also a fundamental change of structure after the HTC process. HC namely was found to be of markedly different structure if compared to pure cellulose (which was found to be very dominant in the sawdust feed), or to pure lignin. Hence, its structure can be described as partly carbonized cellulose. The presence of non-aromatic double bonds, as well as oxygen, might cause an increased chemical reactivity of HC if compared to CB.SEM and TEM analyses of filler dispersion were performed on unvulcanized and vulcanized rubber composite samples. The latter were prepared following a commercial recipe by mixing natural rubber with additives, including a combined filler phase composed of CB and HC. The weight loading of the combined fillers in the rubber composites was kept constant at 50 phr (31.45 wt.%), while the ratio CB/HC varied from 10 CB/40 HC to 50 CB/0 HC. At the finer scale (TEM), both fillers were found to be evenly distributed. At the ‘global scale’ observed by SEM, occasional large particles and/or agglomerates of HC were noted at the highest HC fractions in the fillers’ mixture.Tests of vulcanization rheology of the rubbers with different CB/HC indicated that the HC filler does not markedly change the overall rheological behavior of the rubbers, but that it influences the cure reaction and also the final crosslinking density. HC, on one hand, accelerated the start of the reaction (reduced induction time = ‘scorch time’), but on the other hand, it slowed down and reduced the extent of the crosslinking itself.To sum up, the results of all the above methods indicated that rubber composites, in which 10–20 phr of CB were replaced by bio-sourced HC (prepared under mild process conditions), might be very promising materials, with properties very similar to rubber filled exclusively by CB. These results show a promise of application in significant tonnage for the hardwood waste, as well as for the hydrothermal carbonization (HTC) method. The chemical effect of the predominantly organic HC bio-filler, which was observed in the present experiments, might be exploited after future optimization in adjusted recipes and cure procedures.A follow-on paper is being published as a continuation (second part) of the present work and is dedicated to the mechanical and thermo-mechanical properties of the prepared rubber composites, their swelling behavior, as well as their chemical stability.

## Data Availability

Not applicable.

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
