# Peer review of "Natural Rubber Composites Using Hydrothermally Carbonized Hardwood Waste Biomass as a Partial Reinforcing Filler- Part I: Structure, Morphology, and Rheological Effects during Vulcanization"

_polymers, 2023, doi:10.3390/polym15051176_

Round 1

Reviewer 1 Report

The manuscript presented by the authors is, in my opinion, very interesting. The idea of partial replacement of carbon black with a new, alternative nanoadditive, especially waste and biocompatible, is certainly in line with the latest trends and achievements in the technology of polymeric materials.

The article is written correctly. The introduction section shows the legitimacy of undertaking the presented research topic and highlights the scientific goals. The methodology is well described, with a large number of research techniques used by the authors in their research. The presented results, thoroughly described and detailed conclusions are also worthy of praise.

My comments are basically suggestions to consider and possibly add:

1. It would be worth describing the origin of saw dust a bit more. Was it dust from a single species of tree? If so which one? One could attempt to characterize this material, such as simple laboratory tests of the percentage of lignin and cellulose in the structure of this dust.

2. Why did the authors not prepare a composition containing hydrochar without the use of carbon black? From the point of view of rubber vulcanization properties, it would be extremely valuable - especially in the context of Part II of this article. If possible, I suggest adding such research results at this stage of manuscript improvement.

Author Response

Reviewer#1:

The manuscript presented by the authors is, in my opinion, very interesting. The idea of partial replacement of carbon black with a new, alternative nanoadditive, especially waste and biocompatible, is certainly in line with the latest trends and achievements in the technology of polymeric materials.

The article is written correctly. The introduction section shows the legitimacy of undertaking the presented research topic and highlights the scientific goals. The methodology is well described, with a large number of research techniques used by the authors in their research. The presented results, thoroughly described and detailed conclusions are also worthy of praise.

Answer: The authors are very grateful for the positive assessment, and for the valuable suggestions which helped to improve the Manuscript and make it more attractive.

 NOTE: the line numbers of the discussed changes are correct, if all changes are set to be displayed in the tracking mode.

Reviewer#1:

My comments are basically suggestions to consider and possibly add:

1.  It would be worth describing the origin of saw dust a bit more. Was it dust from a single species of tree? If so which one?

One could attempt to characterize this material, such as simple laboratory tests of the percentage of lignin and cellulose in the structure of this dust.

Answer: The mentioned missing information indeed is interesting.

The hardwood sawdust used as starting material in this work originated from the oak tree.

The content of  cellulose and lignin already was calculated from the elemental analysis  in the original Manuscript (presently line 361 (now bold format): cellulose content: 91 wt.%, lignin content: 9 wt.%) but was only briefly mentioned in the Discussion (l. 361, or Scheme 1 on line 262).

In the revised text, the missing / neglected information about the source material is now more systematically mentioned: on l. 122–123 (Materials), 257–259 (Discussion: Introduction of the filler), 389–393 (13C-NMR analyses), 449–450 (FTIR analyses), 637–638 (Conclusions).

Reviewer#1:

2. Why did the authors not prepare a composition containing hydrochar without the use of carbon black? From the point of view of rubber vulcanization properties, it would be extremely valuable - especially in the context of Part II of this article. If possible, I suggest adding such research results at this stage of manuscript improvement.

 Answer: This question touches an interesting aspect, which unfortunately was not commented in the original Manuscript, but ought to be. Experiments concerned with the preparation of the rubber samples prior to the submission of this work led the authors to the experience, that the rubber samples reinforced exclusively with the prepared hydrochar (and prepared according to the standard procedure described in the Manuscript) did not undergo proper vulcanization. The resulting rubber products systematically displayed distinctly unsatisfactory end-use properties (mechanical, thermal, etc.). Similar experience concerning various types of biochars also can be found upon detailed literature review. For the mentioned reasons, the academically interesting composition filled exclusively by biochar was discarded from the series of specimens to be investigated in detail in this work, and we focused only on the better-performing rubber composites containing at least some carbon black (as the latter was important for successful vulcanization).

It seems that in order to use hydrochar as the exclusive (or dominant) filler, an additional physico-chemical treatment of this filler would be necessary, as well as an optimization of the vulcanization additives – this topic will be the subject of our future research.

In order to address the Question of Reviewer#1, the authors inserted comments explaining why the samples filled only with hydrochar were discarded (l. 143–146 (Materials), and 464–472 (section “3.2. Nanocomposites rubber / CB / hydrochar: basic properties”)). This will be done also in Part II of the article.

Reviewer 2 Report

Review Report Manuscript polymers-2235362

Title:   Natural Rubber Composites Using Hydrothermally Carbonized Hardwood Waste Biomass as a Partial Reinforcing Filler. Part I: Structure, Morphology and Rheological Effects During Vulcanization

In this manuscript which is the first part of a two parts paper, the authors describe the preparation of hardwood sawdust-based hydrothermally carbonized “hydrochar” (HC) as a perspective reinforcing filler in order to partially replace traditional reinforcing carbon black (CB) in Natural Rubber (NR) composites. The demand on renewable raw materials and resources for the rubber industry is the motivation for the presented study.

In this respect the authors claim, that they derived a novel biomass-based filler, which is however quite contradicting having a view on the widely established scientific literature on the use of HC (and also other biomass derived fillers such as lignin or cellulose) as reinforcing fillers for rubber composites. In their literature survey the authors address some of the relevant papers, but lack also others which are published partially in junction with industrialized HC-products, such as the PDEng-thesis of Priyanka Sekar “Design of a Bio-Based Filler System for Tire Treads” of the University of Twente published on 18th June 2020 already and supported by SunCoal Industries GmbH. So, the usage of the term “novel” shall be prevented.

Nevertheless, the authors conducted a very thorough investigation and analysis of the HC filler particle characteristics, their chemical and physical properties, which are analyzed by a set of appropriate techniques including electron microscopy (SEM & TEM), solid state NMR, FTIR and BET-measurements. The authors found and present the expected differences between a N330 CB and the HC in terms of chemical composition, particle size and particle shape as well as fractional dimension, and also in terms of BET-surface, which was nonetheless much more comparable to that of CB (77.8 m2/g for CB vs 21.4 m2/g for HC). However, the comment in lines 282 and 283 on page 7 “….indicates that HC is highly porous, which partly compensates for the small area of the ‘boundary surfaces’ of the large HC grains.” does not really provide any conclusion about the reinforcing capability of the HC vise versa CB, since the filler surface accessible for the rubber molecules to form a strong interaction with the filler surface is usually not only characterized by the BET, but the CTAB surface too, or their ratio which compares internal vs. external surface area. So the presence of a lot of internal micropores is not necessarly a good indication for high reinforcement capability in a high-molecular weight rubber polymer. The authors should address this point in the discussion on the reinforcement capabilities.

The authors prepared five NR-based formulations with a total filler content of 50 phr and varied the CB/HC-ratio form 1:4, 2:3, 3:2, 4:1 and 5:0 in terms of weight. Due to the quite different densities of CB and HC, it would have been helpful to provide also an overview of the respective volume fractions of either of the fillers, because this is an essential information in order to discuss reinforcement capability, curing curves, and mechanical properties (as to be presented in Part 2 of the current paper) of the vulcanizates. The authors used a standard sulfur curing system. In line 135 they name a part of it “stearin”, however it should be rather the term “stearic acid” be used instead.

The compounds were prepared in a laboratory internal mixer according to a procedure presented in a previous publication of the authors (Ref. 29). I have to mark here, that no compound or mixer temperatures are given, so an eventual problem with premature scorch during mixing due to the presence of “highly reactive HC” – which is later on discussed in the vulcameter test section could not be evaluated. The vulcanization behavior was characterized at 150°C by moving die rheometer with the expectable finding that increasing HC-content leads to reduced torque and to the almost disappearance of a scorch safety time. The reasons are briefly given and mostly attributed to the different kinetics and yield of the crosslink reaction, leading to the final torque in the vulcanized state. However, the totally different reinforcing capability of either of the fillers (as it is addressed in the uncured state) is not explicitly mentioned, but should be reflected in the mechanical properties as well (which are to be shown in part 2). But the drastically reduced scorch safety time might be critical for any practical use of the developed CB/HC-NR-formulations, or requires some countermeasures to make the material processable. At least the authors should comment on this issue.

Concluding remark

In general, the work presents a useful and thorough characterization of partially HC-reinforced NR-composites.

In the present form the manuscript should be subject to minor revision, addressing the points I have made in my report above.

Author Response

Reviewer#2:

In this manuscript which is the first part of a two parts paper, the authors describe the preparation of hardwood sawdust-based hydrothermally carbonized “hydrochar” (HC) as a perspective reinforcing filler in order to partially replace traditional reinforcing carbon black (CB) in Natural Rubber (NR) composites. The demand on renewable raw materials and resources for the rubber industry is the motivation for the presented study.

In this respect the authors claim, that they derived a novel biomass-based filler, which is however quite contradicting having a view on the widely established scientific literature on the use of HC (and also other biomass derived fillers such as lignin or cellulose) as reinforcing fillers for rubber composites. In their literature survey the authors address some of the relevant papers, but lack also others which are published partially in junction with industrialized HC-products, such as the PDEng-thesis of Priyanka Sekar “Design of a Bio-Based Filler System for Tire Treads” of the University of Twente published on 18th June 2020 already and supported by SunCoal Industries GmbH. So, the usage of the term “novel” shall be prevented.

Answer: The authors apologize for overlooking the mentioned literature, which was not intentional. The authors added to the Introduction the mentioned PDEng Thesis dedicated to hydrochar filler in tires which was obtained from lignin, namely as the new reference 23 (line 71). Another recent citation about hydrochars from cellulose, hemicellulose and soybean proteins also was added as the new reference 24 (equally on line 71). Additionally, we replaced the term “novel” (filler) with the more appropriate “new-generation” (filler) throughout the Manuscript.

 NOTE: the line numbers are correct, if all changes are set to be displayed in the tracking mode.

Reviewer#2:

Nevertheless, the authors conducted a very thorough investigation and analysis of the HC filler particle characteristics, their chemical and physical properties, which are analyzed by a set of appropriate techniques including electron microscopy (SEM & TEM), solid state NMR, FTIR and BET-measurements. The authors found and present the expected differences between a N330 CB and the HC in terms of chemical composition, particle size and particle shape as well as fractional dimension, and also in terms of BET-surface, which was nonetheless much more comparable to that of CB (77.8 m2/g for CB vs 21.4 m2/g for HC).

However, the comment in lines 282 and 283 on page 7 “….indicates that HC is highly porous, which partly compensates for the small area of the ‘boundary surfaces’ of the large HC grains.” does not really provide any conclusion about the reinforcing capability of the HC vise versa CB, since the filler surface accessible for the rubber molecules to form a strong interaction with the filler surface is usually not only characterized by the BET, but the CTAB surface too, or their ratio which compares internal vs. external surface area. So the presence of a lot of internal micropores is not necessarily a good indication for high reinforcement capability in a high-molecular weight rubber polymer. The authors should address this point in the discussion on the reinforcement capabilities.

Answer: The authors are grateful for this comment. We significantly revised the discussion of the specific surface area of the fillers. The external surface area actually already was estimated by calculation from particle sizes and densities in the original work in the Supplementary Information File, but only briefly and indirectly mentioned in the discussion. In the revised Manuscript, the calculated theoretical values of the external surface area are mentioned in the Discussion and their importance for the reinforcing capability is stressed. The problematic statement that porosity “partly compensates for the small area of the ‘boundary surfaces’ of the large HC grains” was removed.

New discussion text is added on lines 300–322.

Reviewer#2:

The authors prepared five NR-based formulations with a total filler content of 50 phr and varied the CB/HC-ratio form 1:4, 2:3, 3:2, 4:1 and 5:0 in terms of weight.

Due to the quite different densities of CB and HC, it would have been helpful to provide also an overview of the respective volume fractions of either of the fillers, because this is an essential information in order to discuss reinforcement capability, curing curves, and mechanical properties (as to be presented in Part 2 of the current paper) of the vulcanizates.

Answer: This is indeed an interesting aspect. The authors calculated the volume fractions of the co-fillers and added them to the revised Table 2 in the in the section “2.3. Composition of the rubber mixtures”. A brief discussion of the increasing volume fraction of the co-fillers at high HC contents was added as the last part of the discussion of specific surface area and reinforcing capability of the fillers (lines 313–322), and also to the discussion of the vulcanization curves (lines 551–554).

Reviewer#2:

The authors used a standard sulfur curing system. In line 135 they name a part of it “stearin”, however it should be rather the term “stearic acid” be used instead.

Answer: The term “stearin” was replaced by “stearic acid” in the revised Manuscript.

Reviewer#2:

The compounds were prepared in a laboratory internal mixer according to a procedure presented in a previous publication of the authors (Ref. 29). I have to mark here, that no compound or mixer temperatures are given, so an eventual problem with premature scorch during mixing due to the presence of “highly reactive HC” – which is later on discussed in the vulcameter test section could not be evaluated.

Answer: The temperature of the mixing process was mentioned in the presently submitted Manuscript in the section “2.4. Mixing and vulcanization procedure” (90 °C) in the Experimental Part. A discussion of possible premature scorch during mixing was absent, however. This interesting aspect is now discussed in the revised Manuscript (lines 580–583), where the mixing temperature also is newly mentioned.

Reviewer#2:

The vulcanization behavior was characterized at 150°C by moving die rheometer with the expectable finding that increasing HC-content leads to reduced torque and to the almost disappearance of a scorch safety time. The reasons are briefly given and mostly attributed to the different kinetics and yield of the crosslink reaction, leading to the final torque in the vulcanized state. However, the totally different reinforcing capability of either of the fillers (as it is addressed in the uncured state) is not explicitly mentioned, but should be reflected in the mechanical properties as well (which are to be shown in part 2).

Answer: The authors are grateful for this valuable suggestion: The effect of the changing reinforcing capability of the fillers’ mixture is now paid higher attention in the revised Manuscript (lines 544–565).

Reviewer#2:

But the drastically reduced scorch safety time might be critical for any practical use of the developed CB/HC-NR-formulations, or requires some countermeasures to make the material processable. At least the authors should comment on this issue.

Answer: The Reviewer is right about this. The necessity of optimization of the vulcanization behavior and of the preservation of the safety scorch time is now better discussed in the revised Manuscript (lines 590–592).

The section about the vulcanization experiments was thoroughly revised.

Reviewer#2:

Concluding remark

In general, the work presents a useful and thorough characterization of partially HC-reinforced NR-composites.

In the present form the manuscript should be subject to minor revision, addressing the points I have made in my report above.

Answer: The authors are very grateful for the positive overall assessment, and for the well-considered valuable suggestions, which helped to significantly improve the Manuscript and make it more attractive.